# *Borrelia burgdorferi* and *Borrelia miyamotoi* seroprevalence in California blood donors

Sharon I. Brummitt[1]*, Anne M. Kjemtrup[2], Danielle J. Harvey[3], Jeannine M. Petersen[4], Christopher Sexton[4], Adam Replogle[4], Andrea E. Packham[5], Evan M. Bloch[6], Alan G. Barbour[7], Peter J. Krause[8], Valerie Green[9], Woutrina A. Smith[1]

1 Department of Medicine and Epidemiology, School of Veterinary Medicine, University of California, Davis, California, United States of America, 2 California Department of Public Health, Sacramento, California, United States of America, 3 Department of Public Health Sciences, School of Medicine, University of California, Davis, California, United States of America, 4 Division of Vector-Borne Diseases, Centers for Disease Control and Prevention, Fort Collins, Colorado, United States of America, 5 Department of Pathology, Microbiology and Immunology, School of Veterinary Medicine, University of California, Davis, California, United States of America, 6 Department of Pathology and Transfusion Medicine, John Hopkins University, Baltimore, Maryland, United States of America, 7 Department of Medicine and Department of Microbiology and Molecular Genetics, University of California Irvine, Irvine, California, United States of America, 8 Department of Epidemiology of Microbial Diseases, Yale School of Public Health, and Yale School of Medicine, New Haven, CT, United States of America, 9 Creative Testing Solutions, Tempe, Arizona, United States of America

* Sibrummitt@ucdavis.edu

**Data Availability Statement:** All relevant data are within the manuscript and its Supporting Information files.

## Abstract

The western blacklegged tick, *Ixodes pacificus*, an important vector in the western United States of two zoonotic spirochetes: *Borrelia burgdorferi* (also called *Borreliella burgdorferi*), causing Lyme disease, and *Borrelia miyamotoi*, causing a relapsing fever-type illness. Human cases of Lyme disease are well-documented in California, with increased risk in the north coastal areas and western slopes of the Sierra Nevada range. Despite the established presence of *B. miyamotoi* in the human-biting *I. pacificus* tick in California, clinical cases with this spirochete have not been well studied. To assess exposure to *B. burgdorferi* and *B. miyamotoi* in California, and to address the hypothesis that *B. miyamotoi* exposure in humans is similar in geographic range to *B. burgdorferi*, 1,700 blood donor sera from California were tested for antibodies to both pathogens. Sampling was from high endemic and low endemic counties for Lyme disease in California. All sera were screened using the C6 ELISA. All C6 positive and equivocal samples and nine randomly chosen C6 negative samples were further analyzed for *B. burgdorferi* antibody using IgG western blot and a modified two ELISA test system and for *B. miyamotoi* antibody using the GlpQ ELISA and *B. miyamotoi* whole cell sonicate western blot. Of the 1,700 samples tested in series, eight tested positive for antibodies to *B. burgdorferi* (0.47%, Exact 95% CI: 0.20, 0.93) and two tested positive for antibodies to *B. miyamotoi* (0.12%, Exact 95% CI: 0.01, 0.42). There was no statistically significant difference in seroprevalence for either pathogen between high and low Lyme disease endemic counties. Our results confirm a low frequency of Lyme disease and an even lower frequency of *B. miyamotoi* exposure among adult blood donors in California; however, our findings reinforce public health messaging that there is risk of infection by these emerging diseases in the state.

**Funding:** This research was supported in part by a crowdfunding platform through experiment.com set up by S.I.B.: https://experiment.com/projects/ticks-carry-more-than-lyme-disease-in-california-are-californians-at-risk. We would like to acknowledge the California Lyme Disease Association for providing matching funds to the experiment.com crowdfunding platform. This research was also supported in part by National Institutes of Health grant AI-100236 to A.G.B, The National Heart, Lung, and Blood Institute of the National Institutes of Health under award number K23HL151826 to E.M.B, and the Gordon and Llura Gund Foundation to P.J.K. The funders had no role in study design, data collection and analysis, decision to publish, or preparation of the manuscript. There was no additional external funding received for this study.

**Competing interests:** The content of this publication is solely the responsibility of the authors and does not necessarily represent the official views of the National Institutes of Health. Dr. Bloch is a member of the U.S. Food and Drug Administration (FDA) Blood Products Advisory Committee. Any views or opinions expressed in this manuscript are Dr. Bloch's and are based on his own scientific expertise and professional judgment; they do not necessarily represent the views of the Blood Products Advisory Committee or the formal position of the FDA and also do not bind or otherwise obligate or commit either the Advisory Committee or the FDA to the views expressed. The views and opinions expressed herein are those of the authors alone and do not represent the official position of the Centers for Disease Control and Prevention.

## Introduction

California is considered a low incidence state for Lyme disease, defined as a state with a disease incidence of <10 confirmed cases/100,000 annually [1]. The varied ecology results in some counties having higher endemnicity for Lyme disease than others [2–4]. The western black-legged tick (*Ixodes pacificus*) is a common human-biting tick in California and is the principle vector for *Borrelia burgdorferi* sensu stricto, also called *Borreliella burgdorferi* and hereinafter referred to as *B. burgdorferi* [5]. *B. burgdorferi* is the causative agent of Lyme disease in humans and animals in North America. Other potentially zoonotic spirochetes have been documented in the western blacklegged tick [6–8], most notably *Borrelia miyamotoi*, an emerging tick-borne pathogen that is in same genus as the agents of relapsing fever. It causes a febrile illness that occasionally may relapse [9, 10].

In California, the prevalence of *B. burgdorferi* is typically higher in nymphal *I. pacificus* ticks (~3–5%) than adult ticks (~1% or less) [11]. The prevalence for *B. miyamotoi* is about the same (~1%) in both of these tick stages [12–16]. *B. miyamotoi* is transmitted both transstadially and transovarially, which means that while the risk of exposure to *B. burgdorferi* is greater than *B. miyamotoi* after exposure to nymphal ticks, *B. miyamotoi* infection may occur after larval, nymphal, or adult tick exposure and thus extends the season of risk [11]. The distribution of *B. miyamotoi* in *I. pacificus* ticks appears to be similar to that of *B. burgdorferi* and is most prevalent in coastal and foothill regions of northern California [11, 17]. Despite ample evidence of *B. miyamotoi* in California ticks, including ticks that were recovered from humans [18], epidemiological information and case descriptions of *B. miyamotoi* infections in humans are lacking in California.

There are a handful of tick-borne relapsing fever cases caused by *Borrelia hermsii* reported each year in California [16]. Although this spirochete is phylogenetically related to *B. miyamotoi* and shares common antigens [19], its ecology is quite distinct. It is vectored by the soft tick, *Ornithodoros hermsi*, which can be found in rodent nests built in cabins and houses in the mountains and foothill regions of the western United States [20, 21]. The possibility of human infections with *B. miyamotoi* in California comes from a recent study that identified seroreactivity to *B. miyamotoi* in an area highly endemic for Lyme disease in northern California. Seroprevalence was 12%– 14% in the study participants who were at high risk for tick-borne disease, although the GlpQ-based serologic assay used to test for *B. miyamotoi* antibody could not differentiate *B. miyamotoi* infection from *B. hermsii* [22]. The authors surmised that ecology and known behaviors of study participants suggested that they were most likely exposed to *B. miyamotoi* rather than *B. hermsii* [22].

Even with a low prevalence of *B. burgdorferi* and *B. miyamotoi*, California residents are still at risk for tick borne relapsing fever. The purpose of the present study was to determine whether *B. miyamotoi* has a broader geographical range in California than previously demonstrated and to compare the seroprevalence of *B. burgdorferi* and *B. miyamotoi* over this larger range. We therefore assessed human exposure to *B. burgdorferi* and *B. miyamotoi* by testing 1,700 blood bank serum samples from both high and low Lyme disease endemic areas in California. As a broad California-based serosurvey, findings from this study should inform public-health messaging as well as future *B. burgdorferi* and *B. miyamotoi* research.

## Material and methods

### Study population

We obtained 1,700 de-identified human sera samples from Creative Testing Solutions (www.mycts.org) consisting of human sera samples from blood banks. These included 941 samples from high endemic Lyme disease counties in California, defined as > 1 case per 100,000

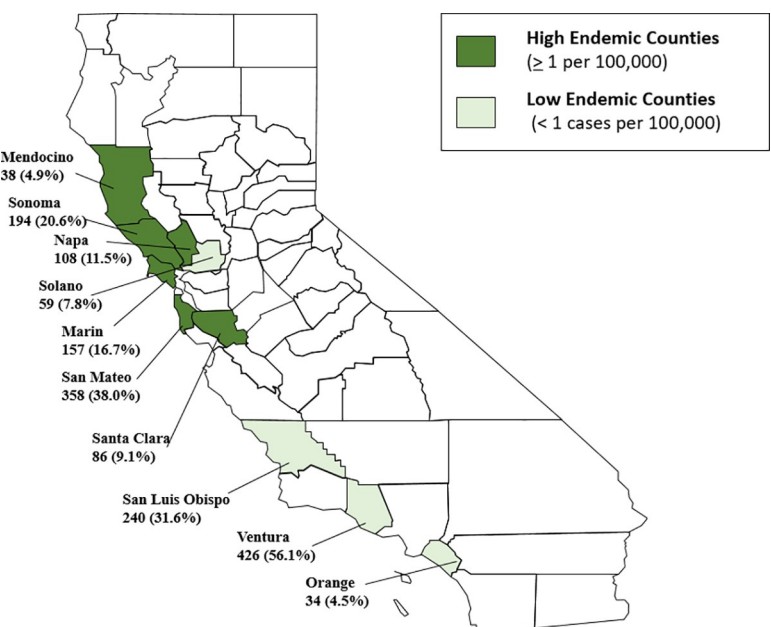

**Fig 1. Counties in California in which samples were collected.** The California counties in dark green represent high endemic counties for Lyme disease and those in light green represent low endemic counties for Lyme disease. Number and percentages of sera samples by county are provided. High endemic Lyme disease counties are defined as those with ≥ 1 case per 100,000 annually and low endemic Lyme disease counties are defined as < 1 case per 100,000 annually [16].

annually and 759 samples from low endemic Lyme disease counties, defined as < 1 case per 100,000 annually [16] (Fig 1). Sample size calculations were performed using Ausvet Epi tools Epidemiological calculators, with sample size calculated based on the following assumptions: (i) an estimated human prevalence of 2% for Lyme disease in high endemic California counties and an estimated human prevalence of 1% for low endemic Lyme disease counties [11, 16]; (ii) a C6 ELISA screening test with a sensitivity of 97% and a specificity ranging from 93% to 99% for antibodies to *B. burgdorferi*; and (iii) a desired precision of 0.012 with an α = .05 and 80% power. Inclusion criteria for blood donor serum samples included the zip code of the blood donor, sera that was non-reactive on screening assay for other infectious diseases including Hepatitis B or C, HIV 1 or 2, HTLV I/II, West Nile virus, Zika virus, and syphilis; and a minimum volume of at least 1ml. Samples were collected from April 2017 through June 2017, corresponding with typical seasonality for Lyme disease cases reported in California [11]. Information associated with the samples included general demographic information such as sex, age, ethnicity, county and zip code of the donor.

## Ethics approval

De-identified serum samples with associated demographic variables for each serum sample were submitted for this study. The study was determined to have exempt status by the Institutional Review Board, University of California Davis (Protocol Number 1090480–1, FWA No: 00004557).

## Sample processing and storage

Creative Testing Solutions shipped frozen one-ml aliquot sera samples on dry ice to the testing facility at the University of California, Davis (UCD). UCD stored all sera samples at -80°C

until thawed in the refrigerator for one to five days until testing. By using the commercial C6 ELISA test as a screening tool, the full set of samples could be initially evaluated for reactivity to *B. burgdorferi* and *B. miyamotoi* since both of these agents react to this antigen [23, 24]. The IgG western blot for *B. burgdorferi* and total Ig GlpQ ELISA and IgG western blot for *B. miyamotoi* were subsequently used to test all C6 positive/equivocal samples, as well as nine randomly selected C6 negative samples, to serve as controls [23, 25–27]. Additionally, a recently FDA approved test system for detection of *B. burgdorferi* antibodies, referred to as the Modified Two Tier Test (MTTT) (Zeus Scientific, New Jersey), was performed in accordance with the manufacturer's guidelines on the C6 positive/equivocal and C6 negative samples [28]. After C6 screening (described below), all C6-positive/equivocal and the randomly chosen negative samples were separated into 250μl aliquots in coded tubes, re-coded to mask county of location, and shipped frozen to the Centers for Disease Control and Prevention (CDC). CDC stored all sera samples in a -80°C freezer until testing (Fig 2).

## Standard two-tiered testing

All sera samples were screened using a C6 Lyme ELISA kit (Immunetics, Boston Massachusetts) performed in accordance with the manufacturer's recommendation. Testing was performed in duplicate for all specimens. A Lyme index value for each sample was calculated by dividing the average of the sample's OD values by the calibrator cutoff value. Positive and negative controls were provided by the manufacturer. Samples positive or equivocal by C6 Lyme ELISA were tested using the second-tier (IgG) Marblot western blot (MarDx Diagnosis, Trinity Biotech, Carlsbad, California) which was performed in accordance with the manufacturer's recommendation. The criteria used in the interpretation of the western blot as either negative or positive was based on the manufacturer's recommendation using established criteria [29].

## Modified two-tiered testing

The *Borrelia* VlsE1/pepC10 IgM/IgG test system (ZEUS Scientific, Branchburg, NJ) was performed on 91 C6 ELISA positive/equivocal samples as well as nine randomly chosen negative samples. All positive and equivocal samples were then tested by the *B. burgdorferi* ELISA whole cell antigen IgG test system (ZEUS Scientific) [30]. Both ELISAs were performed in accordance with the manufacturer's recommendation [31].

## GlpQ ELISA

*B. miyamotoi* recombinant his-tagged GlpQ antigen (1 μg/well) was bound to 96 well plates as described [24] with the following modifications. Peroxidase conjugated goat anti-human IgA +IgG+IgM (H&L) (1:2500) and SureBlue TMB peroxidase substrate (SeraCare, Milford, Massachusetts) were utilized to detect bound antibody. The positive cutoff was set at three standard deviations above the mean absorbance of sera from four negative controls (healthy persons). A positive control from a *B. miyamotoi* PCR positive patient was included in each run. Absorbance was read at 450 nm.

## *B. miyamotoi* western blot

The *B. miyamotoi* western blot strips were produced using *B. miyamotoi* strain CT13-2396, which was originally isolated from *I. scapularis* collected in Connecticut; NCBI accession number: PRJNA310783. Strain CT13-2396 was grown in BSK-R medium and harvested by centrifuging at 10,000(g) for 10 minutes at 4°C. The resulting cell pellet was frozen, thawed, and re-suspended in TE buffer (Fisher Scientific, Pittsburgh, PA), sonicated, and diluted to a

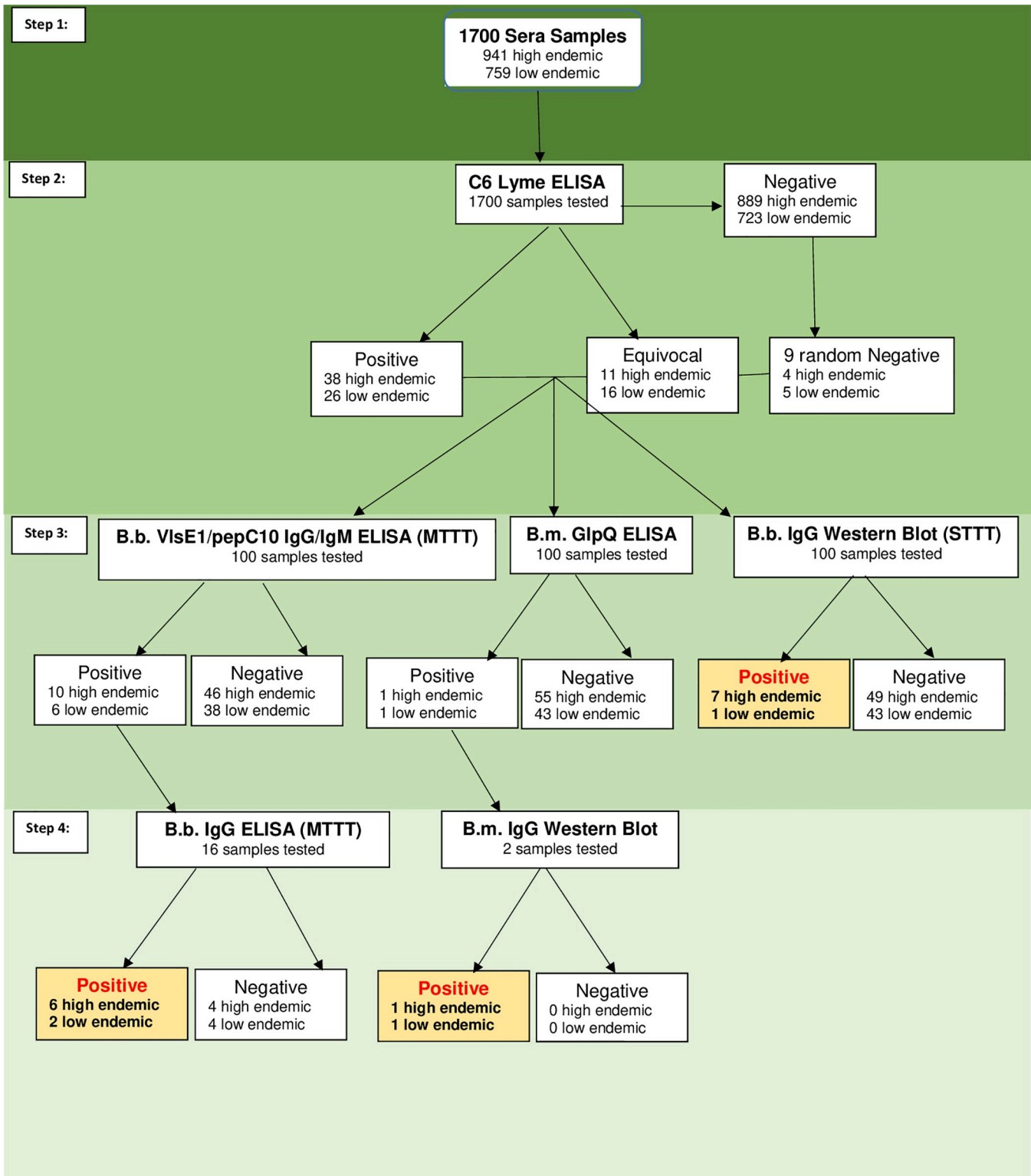

**Fig 2. Flowchart of testing strategies.** Testing strategy followed for the 1,700 blood donors from California. Each step of the diagram details the testing strategy and the number of samples tested.

final protein concentration of 2.0 mg/ml. The sonicate was mixed with sample buffer with DTT (Bio-Rad, Inverness, CA), heated at 95˚C heat block for 10 min, resolved on 12.5% SDS-PAGE gels for 180 minutes at 70mAmps, and soaked in tris/glycine buffer (Bio-Rad) with 20% methanol for 30 minutes at 4˚C. Separated proteins were transferred for 30 minutes at 25 volts to a 0.2 μm nitrocellulose membrane using a Trans-Blot® SD Semi-Dry Transfer Cell (Bio-Rad). Membranes were soaked overnight in 1% milk (Bio-Rad) and tris-buffered saline and Tween™ 20 (TBST) (Fisher Scientific), dried and then cut into 3 mm strips and stored at 4˚C. To perform western blotting, strips were re-hydrated in 1% milk and TBST (blocking buffer), then incubated with sera at a final concentration of 1:200 in blocking buffer for 30 min. Strips were then washed and incubated for 15 min in blocking buffer and phosphatase-labeled goat anti human IgG (H+ L) conjugate (KPL, Gaithersburg, MD) added at a concentration of 1:10,000, followed by a final wash series. Strips were developed using BCIP/NBT phosphatase substrate (KPL, Gaithersburg, MD). A control strip using a monoclonal GlpQ antibody was included as a locator for the GlpQ antigen; it was processed identically, with the exception of using anti mouse IgG (H+ L) conjugate. *B. miyamotoi* and *B. burgdorferi* human sera were included as positive and negative controls, respectively.

## Statistical analysis

Statistical analysis of *B. burgdorferi* and *B. miyamotoi* results were performed separately. The outcome variable for each analysis was the serum sample as positive, equivocal, or negative for the pathogen. The exposure variable was blood donor's residence in high or low Lyme disease endemic counties, as defined above. Point prevalence (seroprevalence) was calculated as the number of people who tested positive for the pathogen over total number of people tested. Exact 95% confidence intervals were constructed for prevalence estimates. Sample t-tests and chi-square tests were used to compare characteristics of those providing samples between the high and low-endemic counties in the overall sample and those samples carried forward to confirmatory testing. A two-tailed Fisher's exact test was performed to compare the percentage of *Borrelia* species–seropositive study participants in the high and low endemic counties. A P value less than 0.05 was considered statistically significant. All analyses were performed using SAS software version 9.4, SAS Institute Inc., Cary, NC, USA.

## Results

### Sample characteristics

There was no significant difference in the distribution of males to females by risk level for the study population. Participants in low Lyme disease endemic counties were on average slightly younger than those in high Lyme disease endemic counties (41 years and 49 years, respectively P<0.001). The majority of participants listed ethnicity as White and non-Hispanic, with a greater percentage of Hispanics in the low endemic counties (20.5%) than in the high endemic counties (6.7%; P<0.001) (Table 1). In the high endemic counties, over two-thirds of the samples were from San Mateo (38.0%) or Sonoma (20.6%) counties, while most of the samples from the low endemic counties were from Ventura (56.1%) and San Luis Obispo (31.6%) counties (Table 2).

### C6 ELISA screen

Of the 1,700 serum samples screened with C6 ELISA, 64 (3.76%) were positive and 27 (1.59%) were equivocal. Of the 941 samples from high endemic counties, 49 (5.2%) total samples were

**Table 1. Demographic characteristics of sera samples and positive Bb STTT and Bm GlpQ.**

| | All Sera Samples | | P-Value | Positive *B. burgdorferi* Standard Two Tiered Testing** | | Positive *Borrelia miyamotoi* GlpQ** | |
| --- | --- | --- | --- | --- | --- | --- | --- |
| | | | | (C6 ELISA + Marblot WB) | | (GlpQ ELISA + Bm WB) | |
| | High Endemic | Low Endemic | | High Endemic | Low Endemic | High Endemic | Low Endemic |
| | (N = 941) | (N = 759) | | | | | |
| Sex | | | | | | | |
| Male | 521 (55.4%) | 420 (55.3%) | 0.7 | 7 (0.74%) | 0 (0.00%) | 0 (0.00%) | 0 (0.00%) |
| Female | 413 (43.9%) | 346 (45.6%) | | 0 (0.00%) | 1 (0.13%) | 1 (0.11%) | 1 (0.13%) |
| Age | | | | | | | |
| Mean Age (SD*) | 49.0 (16.9) | 41.1 (19.7) | < **0.001** | 65.3 yrs (6.5) | 63 yrs (N/A) | 17 yrs (N/A) | 32 yrs (N/A) |
| Range | 16yrs to 84yrs | 16yrs to 84yrs | | 56 yrs—75 yrs | N/A | N/A | N/A |
| Ethnicity | | | | | | | |
| Non-Hispanic | 878 (93.3%) | 603 (79.4%) | < **0.001** | 7 (0.74%) | 1 (0.13%) | 1 (0.11%) | 0 (0.00%) |
| Hispanic | 63 (6.7%) | 156 (20.5%) | | 0 (0.00%) | 0 (0.00%) | 0 (0.00%) | 1 (0.13%) |

* Standard deviation not calculated for (n = 1)

** Percentages calculated out of all sera samples in high or low endemic areas

positive (n = 38) or equivocal (n = 11) for *B. burgdorferi* antibody compared to 42 (5.5%; 26 positive and 16 equivocal) of the 759 samples from low endemic counties.

Ninety-one samples that were C6 ELISA positive/equivocal and nine randomly selected negative samples had additional testing. A little more than half (58%) of the 91 samples that were positive or equivocal were from male individuals. The mean age for males was 42.6 years (range 16 years to 80 years) while the mean age of females was 49.6 years (range 16 years to 77 years). Most individuals with positive or equivocal samples were non-Hispanic (87%). Ethnicity (high endemic: Non-Hispanic (93.9%); low endemic: Non-Hispanic (78.6%); P-value = 0.058), sex (high endemic: male 55.1%; low endemic: male 61.9%; P-value = 0.5) and age (high endemic:

**Table 2. Testing results of sera samples for *B. burgdorferi* and *B. miyamotoi* by county.**

| Counties where Sera was Sampled | C6 Lyme ELISA/Screening Test** | Bb Standard Two Tier Testing*** | *Borrelia miyamotoi* GlpQ*** |
| --- | --- | --- | --- |
| N | n (%*) | n (%*) | n (%*) |
| **High Endemic Counties** | | | |
| Marin (n = 157) | 12 (7.64%) | 2 (1.27%) | 0 (0.00%) |
| Mendocino (n = 38) | 3 (7.89%) | 0 (0.00%) | 1 (2.63%) |
| Napa (n = 108) | 3 (2.78%) | 1 (0.93%) | 0 (0.00%) |
| San Mateo (n = 358) | 19 (5.31%) | 3 (0.84%) | 0 (0.00%) |
| Santa Clara (n = 86) | 3 (3.49%) | 0 (0.00%) | 0 (0.00%) |
| Sonoma (n = 194) | 9 (4.64%) | 1 (0.51%) | 0 (0.00%) |
| **Low Endemic Counties** | | | |
| Orange (n = 34) | 0 (0.00%) | 0 (0.00%) | 0 (0.00%) |
| San Luis Obispo (n = 240) | 16 (6.67%) | 1 (0.42%) | 0 (0.00%) |
| Solano (n = 59) | 3 (5.08%) | 0 (0.00%) | 0 (0.00%) |
| Ventura (n = 426) | 23 (5.40%) | 0 (0.00%) | 1 (0.23%) |

## Positive test

* County percent is based upon the number of samples tested from each county.

** County percent include both positive and equivocal results

***County percent include only positive test results

mean = 49 years, SD = 16.6; low endemic: mean = 42 years, SD = 20.9; P-value = 0.07), of those with positive or equivocal samples did not differ between high and low endemic counties.

Of the nine randomly selected negative samples, two thirds were from female subjects. The mean age was 40.6 years for males (range 16 years to 72 years) and 35.7 years (range 25 years to 60 years) for females. Seven of the nine (78%) were non-Hispanic. Four of the nine (44%) selected negative of the samples were from high endemic counties while 5 of the 9 (56%) of samples were from low endemic counties.

## *B. miyamotoi* GlpQ seroreactivity

Two of the 1,700 samples had detectable antibodies against *B. miyamotoi* (0.12%, Exact 95% CI: 0.01%, 0.42%). Both samples tested positive by C6 ELISA, GlpQ ELISA and *B. miyamotoi* whole cell western blot. Both samples were negative on the IgG western blot for *B. burgdorferi* (Fig 3, Table 3). None of the nine randomly selected C6 seronegative samples were seropositive for antibodies against *B. miyamotoi* by the GlpQ ELISA. Seroprevalence was (2.63%, Exact 95% CI: 0.7, 13.81) among Mendocino County residents and (0.23%, Exact 95% CI: 0.01, 1.30) among Ventura County residents. Both seropositive samples were from females with respective ages of 17 years and 32 years (Table 1).

## *B. burgdorferi* IgG western blot

Eight of 1,700 samples had detectable antibodies against *B. burgdorferi* (0.47%, Exact 95% CI: 0.20, 0.93). Of the eight sera samples that were positive by the C6 ELISA and *B. burgdorferi*

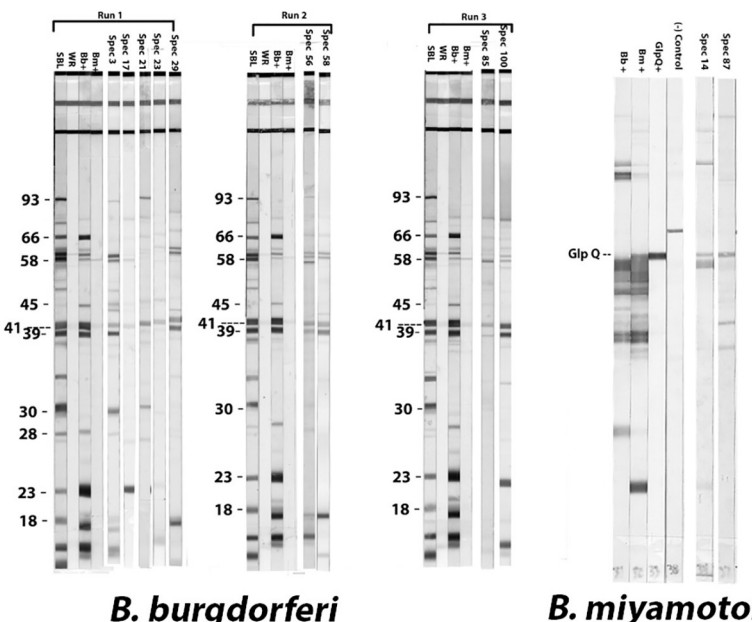

**Fig 3. *B. burgdorferi* and *B. miyamotoi* western blot seroreactivity in blood donors from higher and lower risk counties for Lyme disease in California.** For *B. burgdorferi*, each run included a serum band locator (SBL) control, which shows reactivity to Lyme diagnostically significant bands at position 93, 66, 58, 45, 41, 39, 30, 28, 23 and 18 kDa and weakly reactive (WR) control, which shows reactivity to the 41 kDa band. Additionally, control serum from persons confirmed positive for *B. burgdorferi* or *B. miyamotoi* were included. Serum samples seropositive for 5 or more bands are shown. For *B. miyamotoi* western blots, control sera from patients confirmed positive for *B. burgdorferi* or *B. miyamotoi* were included along with a negative control serum. A monoclonal antibody to GlpQ was also included as a locator for the GlpQ antigen. The two samples with seroreactivity to GlpQ are shown.

**Table 3. Antibody testing results of human samples.**

| Study Subject | *B. burgdorferi* (Standard Two Tier Testing) | | | | | *B. burgdorferi* (Modified Two Tier Testing) | | | | | *B. miyamotoi* (GlpQ Testing) | | | |
|---|---|---|---|---|---|---|---|---|---|---|---|---|---|---|
| Sample Number | C6 ELISA | | Marblot WB (IgG)* | | Interpretation | Zeus vslE1/pepC10 | | Zeus whole cell IgG | | Interpretation | GlpQ ELISA | | *B. miyamotoi* Western Blot | |
| | Index Value | Results | Pos Band | Results | | Index Value | Results | Index Value | Results | | OD value | > pos cutoff | GlpQ Band | Result |
| **High Endemic Counties** | | | | | | | | | | | | | | |
| Sample 3 | **6.660** | Positive | 7 | Positive | Positive | **4.468** | Positive | 3.362 | Positive | Positive | 1.210 | 2.799 | No | Negative |
| Sample 17 | **1.926** | Positive | 5 | **Positive** | **Positive** | 0.269 | Negative | N/A | N/A | Negative | 1.471 | 2.799 | No | Negative |
| Sample 21 | **4.510** | Positive | 6 | **Positive** | **Positive** | 2.337 | Positive | 1.428 | Positive | Positive | 1.278 | 2.799 | No | Negative |
| Sample 29 | **1.468** | Positive | 5 | **Positive** | **Positive** | 1.448 | Positive | 2.920 | Positive | Positive | 0.953 | 2.145 | No | Negative |
| Sample 56 | **3.783** | Positive | 6 | **Positive** | **Positive** | 1.295 | Positive | 2.799 | Positive | Positive | 0.589 | 2.234 | No | Negative |
| Sample 58 | **1.156** | Positive | 5 | **Positive** | **Positive** | 0.991 | Equivocal | 2.083 | Positive | Positive | 0.622 | 2.234 | No | Negative |
| Sample 87 | **1.190** | Positive | 4 | Negative | Negative | 0.254 | Negative | N/A | N/A | Negative | **2.493** | **2.246** | Yes | **Positive** |
| Sample 100 | **1.082** | Equivocal | 5 | **Positive** | **Positive** | 1.366 | Positive | 2.365 | Positive | Positive | 1.145 | 2.246 | No | Negative |
| **Low Endemic Counties** | | | | | | | | | | | | | | |
| Sample 14 | **0.990** | Equivocal | 2 | Negative | Negative | 0.675 | Negative | N/A | N/A | Negative | **3.094** | **2.799** | Yes | **Positive** |
| Sample 23 | **0.957** | Equivocal | 6 | **Positive** | **Positive** | 0.327 | Negative | N/A | N/A | Negative | 2.015 | 2.799 | No | Negative |
| Sample 59 | **3.540** | Positive | 2 | Negative | Negative | **1.604** | Positive | 0.965 | Equivocal | Positive | 1.591 | 2.234 | No | Negative |
| Sample 79 | >7.00 | Positive | 3 | Negative | Negative | **6.437** | Positive | 2.518 | Positive | Positive | 0.360 | 1.666 | No | Negative |
| Sample 85 | 0.220 | Negative | 5 | **Positive** | Negative | 0.225 | Negative | N/A | N/A | Negative | 0.865 | 1.666 | No | Negative |

* Marblot Western Blot—A positive sample is ≥ 5 bands out of 10 bands (All results were visually read).

IgG western blot (STTT), seven (0.74%, Exact 95% CI: 0.30, 1.53) were from high Lyme disease endemic counties and one positive sample (0.13%, Exact 95% CI: 0.00, 0.73) was from a low Lyme disease endemic county (P = 0.08) (Table 1). Marin County had an overall seroprevalence of 1.3% (Exact 95% CI: 0.15, 4.53), Napa county had an overall seroprevalence of 0.93% (Exact 95% CI: 0.02, 5.05), San Mateo County had an overall seroprevalence of 0.84% (Exact 95% CI: 0.17, 2.43), and Sonoma County had an overall seroprevalence of 0.52% (Exact 95% CI: 0.01, 2.84). San Luis Obispo had an overall seroprevalence (0.4%) (Exact 95% CI: 0.01, 2.30) (Table 2).

## Modified two-tiered testing

Eight of the 91 C6 ELISA positive/equivocal sera were positive by modified two-tiered testing (MTTT) (Table 3). Six of the 91 C6 ELISA positive/equivocal sera positive by both STTT and MTTT were from a high endemic county and two of 91 C6 ELISA positive/equivocal sera positive by MTTT only were from a low endemic county. There was little agreement in STTT and MTTT positivity among C6 ELISA positive/equivocal sera from a low endemic county. Seven

of the 91 C6 ELISA positive/equivocal sera positive by MTTT were male with mean age of 65 years (range 56 years to 75 years) and the age of the female was 16 years.

## Discussion

This study represents the largest serosurvey across a broad geographic area in California estimating human exposure to *B. miyamotoi* and *B. burgdorferi*. We found higher *B. burgdorferi* antibody estimates from higher risk Lyme disease endemic areas in northern California but similar *B. miyamotoi* antibody estimates from high and low risk Lyme disease endemic areas in the state. Our large sample size of 1,700 and broad geographic expanse increases the confidence of our seroprevalence estimate. Although the overall risk of human acquisition of either pathogen is lower in California compared with those in high risk Lyme disease endemic areas in the Northeast and northern Midwest, endemic areas are shifting with climate and human habitat changes that alter the epidemiology of these infections [32, 33]. Our data align with previous research that demonstrated the C6 Lyme ELISA test can detect seroreactivity to both *B. burgdorferi* and *B. miyamotoi* because the C6 peptide sequence in the C6 ELISA test kit found in *B. burgdorferi* is very similar to the relapsing fever *Borrelia* variable large protein (Vlp) sequence, including Vlp 15/16 of *B. miyamotoi* [23, 34]. The cross-reactive antibodies against the C6 peptide occurs in 90% of patients with *B. miyamotoi* disease [34]. The C6 ELISA test can also detect antibodies to all of the major pathogenic European *Borrelia* species: *B. afzelii*, *B. garinii*, and *B. burgdorferi* [35].

Only two of the 1,700 serum samples were positive for both C6 and GlpQ antibodies (0.12%, Exact 95% CI: 0.01, 0.42). This result suggests that these two persons had prior *B. miyamotoi* infection, given that the *glpQ* gene is not present in Lyme spirochetes [23, 24]. With so few *B. miyamotoi* seropositive samples, we had insufficient power to detect a difference between high and low Lyme disease endemic counties. Overall in California, the risk of exposure to *B. miyamotoi* is relatively low. However, there may be ecologic foci of exposure risk, similar to what is seen for Lyme disease in California [2]. For example, the *B. miyamotoi* seroprevalence in Mendocino County (2.6%) is similar to that noted in a previous study [22]. which documented seroprevalence values of 1.98% and 6.93% over several years in a Mendocino community. The population studied was at high risk of tick-borne disease because of well-documented *I. pacificus* tick exposure [22].

There is a potential for cross reactivity to the *B. miyamotoi* GlpQ antigen in persons with prior exposure to *B. hermsii* [36, 37]. *Borrelia hermsii* is the primary etiologic agent of tick-borne relapsing fever (TBRF) in the western United States and transmitted by the argasid (soft) tick *Ornithodoros hermsi* [38, 39] with about 2 to 20 cases per year in California [40]. The primary hosts for these ticks are rodents [38, 39]. The range of the vector that carries *B. hermsii* are typically found at higher elevations (914 meters to 2743 meters) from the southern Cascades to the Eastern Sierra Nevada mountain range down to the Southern California Mountains [38]. Ventura County's proximity to *B. hermsii* endemic areas (within 120 km), and the fact that no *I. pacificus* ticks have tested positive for *B. miyamotoi* from Ventura County [16, 20, 40, 41], suggest that we cannot rule out that the *B. miyamotoi* seropositive sample from Ventura County represents seropositivity to *B. hermsii*. Mendocino County, by contrast, has had *I. pacificus* ticks with documented *B. miyamotoi* infection and is geographically distant from *O. hermsii* distribution [22], supporting the *glpQ* -positive sample more likely to represent *B. miyamotoi* exposure. Both GlpQ positive samples for *B. miyamotoi*, not only can represent cross reactivity with *B. hermsii*, but the infections may have been acquired outside the county or state. Although *B. miyamotoi* has a low seroprevalence in California, our study findings are consistent with a previous study that humans in California are exposed to *B. miyamotoi* [22].

The overall seroprevalence for *B. burgdorferi* from both high and low Lyme disease endemic counties in California for Lyme disease was 0.47% (Exact 95% CI: 0.20, 0.93). The STTT, utilizing the C6 ELISA and IgG western blot, is more stringent and specific then the IgM western blot [42]. Since blood bank donors are generally healthier than the general population [43, 44] and the incidence of Lyme disease in California is about 0.2 cases per 100,000 population [16], a *B. burgdorferi* seroprevalence of less than 0.5% in California appears to be a reasonable estimate.

For California, a local approach to estimating risk is important for public health communication, given the well-documented non-uniform exposure due to local ecological influences [2, 11, 17, 45]. Few studies are available from California measuring *B. burgdorferi* seroprevalence in specific communities. In one study in Sonoma County, 1.4% of a small community were found to be seropositive for *B. burgdorferi*, an estimate within our confidence level estimates for that county, while in the same study, no samples from a blood bank collection from Sacramento County tested positive [46]. Although we did not find a statistically significant difference in seroprevalence between high and low Lyme disease endemic counties, all but one of our *B. burgdorferi* positive samples came from high endemic counties. Our study provides an updated estimate of *B. burgdorferi* exposure in a broad geographic area of California and helps demonstrate that risk for Lyme disease is geographically diverse. A more recent study found that 3.2% of residents in high endemic counties for Lyme disease in northern California may have been exposed to *B. burgdorferi* [47] as measured by the C6 ELISA alone. This estimate is within our confidence level estimates for C6 positive/equivocal samples. However, it is important that our C6 positive or equivocal samples were followed by testing with a more specific IgG Lyme western blot assay to avoid over estimation of *B. burgdorferi* exposure and to differentiate *B. miyamotoi* exposure. The increased sensitivity and decreased specificity of the C6 ELISA could over-estimate true exposure if used as a stand-alone test for estimating seropositivity, particularly in low incident states [48, 49] and should be used in conjunction with second tier assays in the recommended STTT format for Lyme disease diagnosis [50, 51].

The MTTT is intended for the qualitative detection of antibodies to *B. burgdorferi* in human serum [30]. With the advent of both the VslE1 and peptide C10 in the first tier ELISA assay, the MTTT assay we utilized is designed to be both sensitive and specific for early and late infection in patients with Lyme disease [30, 52]. The C6 peptide derived from VlsE1 does not bind to IgM well, and therefore the C6 ELISA is not ideal for detecting early cases of Lyme disease [48, 53]. On the other hand, an IgM response is generated early in disease in response to the pepC10 protein [52]. Among the 91 samples seropositive or equivocal by the C6 ELISA, the overall seropositivity was the same for MTTT and STTT. However, minor discrepancies were noted: Samples 59 and 79 were negative by STTT but positive by MTTT and Samples 17 and 23 were positive by STTT but negative by MTTT. Samples 59 and 79 had high Lyme index values by C6 ELISA but were negative on the Lyme disease western blot (IgG), possibly indicating that both samples were taken early after infection or the person had been treated early after illness onset and did not produce an expanded IgG response detectable by the Lyme disease western blot criteria. In contrast, samples 17 and 23 had lower index values by C6 ELISA and were positive by western blot, possibly indicating an old infection, with waning pepC10 antibody response that could not be picked up by the MTTT.

There are some limitations to this study. The necessary deidentified nature of human serum from blood bank samples precludes analysis of potential risk factors such as travel to *B. burgdorferi*, *B. miyamotoi* or *B. hermsii* endemic areas or degree of tick exposure. Human antibodies to the C6 peptide for *B. burgdorferi* wane after two years following untreated *B. burgdorferi* infection and probably more rapidly if treated [54]. The sensitivity of C6 for detecting *B. miyamotoi* infection and the longevity of the C6 antibody response in *B. miyamotoi* infection

is also not known. Human IgG antibody response dynamics to *B. miyamotoi* are still unknown [55]. Lack of exposure information decreases the overall prior probability of having the disease in relation to diagnostic testing. Only nine negative samples were chosen due to resource limitations of second tier testing. Though the samples could have been matched by gender, given the small number of negative samples available, the authors chose to balance by endemicity status. Since there was no difference in seropositivity between males and females, we felt that the 2/3 bias towards females in the negative samples did not compromise the study. Sampling sera from blood donors may decrease the generalizability of the results to the general population [43].

## Conclusions

Our study demonstrates that California residents are at risk for infection by the emerging *I. pacificus*-transmitted pathogens *B. miyamotoi* and *B. burgdorferi*, although they are at relatively low risk of these infections even in the most highly endemic counties for Lyme disease. Standard two-tier *B. burgdorferi* testing and GlpQ serologic testing for *B. miyamotoi* on C6 positive or equivocal samples determined specific reactivities to these agents. Among samples testing C6 positive/equivocal, the STTT and MTTT performed fairly consistently even in this population with no known exposure history and in a state with low endemicity for Lyme disease. Further investigation of risk mapping related to geography and habitat type are needed for *B. miyamotoi*.

## Supporting information

**S1 Raw images.**
(PDF)

## Acknowledgments

We would like to acknowledge Dr. Christopher Barker from the University of California, Davis for his helpful input and review on earlier drafts of this manuscript as well as his support as a member of my dissertation committee. We would like to also acknowledge the California Lyme Disease Association for their matching funds with the experiment.com crowdfunding platform. We would also like to thank Ruwini K. Rupasinghe, Phd Student in Epidemiology at UC Davis for her help in C6 Testing of all 1700 blood samples.

## Author Contributions

**Conceptualization:** Sharon I. Brummitt, Anne M. Kjemtrup, Jeannine M. Petersen, Evan M. Bloch, Alan G. Barbour, Peter J. Krause, Woutrina A. Smith.

**Data curation:** Sharon I. Brummitt, Anne M. Kjemtrup, Andrea E. Packham.

**Formal analysis:** Sharon I. Brummitt, Danielle J. Harvey.

**Funding acquisition:** Sharon I. Brummitt, Woutrina A. Smith.

**Investigation:** Sharon I. Brummitt, Andrea E. Packham, Valerie Green.

**Methodology:** Sharon I. Brummitt, Anne M. Kjemtrup, Danielle J. Harvey, Jeannine M. Petersen, Christopher Sexton, Adam Replogle, Andrea E. Packham, Alan G. Barbour, Peter J. Krause.

**Project administration:** Sharon I. Brummitt, Woutrina A. Smith.

**Resources:** Evan M. Bloch, Alan G. Barbour, Valerie Green, Woutrina A. Smith.

**Supervision:** Anne M. Kjemtrup, Danielle J. Harvey, Jeannine M. Petersen, Woutrina A. Smith.

**Writing – original draft:** Sharon I. Brummitt, Anne M. Kjemtrup.

**Writing – review & editing:** Sharon I. Brummitt, Anne M. Kjemtrup, Danielle J. Harvey, Jeannine M. Petersen, Christopher Sexton, Adam Replogle, Andrea E. Packham, Evan M. Bloch, Alan G. Barbour, Peter J. Krause, Valerie Green, Woutrina A. Smith.

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
