## [Decision Letter · Decision Letter 0]

10 Sep 2020

PONE-D-20-23819

Borrelia burgdorferi and Borrelia miyamotoi Seroprevalence in California Blood Donors

PLOS ONE

Dear Dr. Brummitt,

Thank you for submitting your manuscript to PLOS ONE. After careful consideration, we feel that it has merit but does not fully meet PLOS ONE’s publication criteria as it currently stands. Therefore, we invite you to submit a revised version of the manuscript that addresses the points raised during the review process.  Again, the reviewers believed the manuscript has interesting data that is appropriate for this journal, but the points raised by the reviewers are stated clearly and we hope you can address these in a timely manner.

We look forward to receiving your revised manuscript.

Kind regards,

R. Mark Wooten, Ph.D.

Academic Editor

PLOS ONE

Journal Requirements:

2. Please move the following paragraph to the Competing Interests statement (from the Financial Disclosure section):

"The content of this publication is solely the responsibility of the authors and does not necessarily represent the official views of the National Institutes of Health. Dr. Bloch is a member of the U.S. Food and Drug Administration (FDA) Blood Products Advisory Committee. Any views or opinions expressed in this manuscript are Dr. Bloch's and are based on his own scientific expertise and professional judgment; they do not necessarily represent the views of the Blood Products Advisory Committee or the formal position of the FDA and also do not bind or otherwise obligate or commit either the Advisory Committee or the FDA to the views expressed. The views and opinions expressed herein are those of the authors alone and do not represent the official position of the Centers for Disease Control and Prevention."

Reviewers' comments:

Reviewer's Responses to Questions

**Comments to the Author**

1. Is the manuscript technically sound, and do the data support the conclusions?

Reviewer #1: Yes

Reviewer #2: Yes

2. Has the statistical analysis been performed appropriately and rigorously? 

Reviewer #1: I Don't Know

Reviewer #2: Yes

3. Have the authors made all data underlying the findings in their manuscript fully available?

Reviewer #1: Yes

Reviewer #2: Yes

4. Is the manuscript presented in an intelligible fashion and written in standard English?

Reviewer #1: Yes

Reviewer #2: Yes

5. Review Comments to the Author

Reviewer #1: Review: Borrelia burgdorferi and Borrelia miyamotoi Seroprevalence in California Blood Donors.

This study assessed the seroprevalence of B. burgdorferi and B. miyamotoi in 1700 blood donors in California. Donors were selected from counties with reported high prevalence of B. burgdorferi infection as well as those with low prevalence. Of the 1700 donors, 91 were positive (or equivocal) by a standalone C6 ELISA assay for B. burgdorferi. When two tier (STTT or MTTT) assay was applied only 8 of the 91 donors were positive for B. burgdorferi antibodies. Two donors were positive for B. miyamotoi antibodies. No major differences in seroprevalence were observed between high and low prevalence areas. In total 0.47% of patients had serological evidence of B. burgdorferi infection and 0.12% had serological evidence of B. miyamotoi infection. The findings demonstrate low prevalence of B. burgdorferi and B. miyamotoi infection in California.

A key strength of this study is the assessment of seroprevalence for both agents at two independent testing sites (UCD and CDC) and the use of 2 testing methodologies (STTT and MTTT) for B. burgdorferi. The results are largely consistent with previous findings. The major comment is regarding the data presentation in the Results, which could be improved.

1) Figure 1 should include the number of samples that came from each county as well as the percentage. Figure Legends should be provided.

2) The 2nd paragraph of the Results (C6 ELISA Screen) provides information on how the 91 patients were selected for downstream analyses, but this information is not presented anywhere. My suggestion is to present this in a form of a Table or a flow chart since all subsequent analyses in the manuscript stem from this selection/screen.

3) In the second paragraph of the C6 ELISA Screen section, the authors discuss the distribution of Hispanics and non-Hispanics among low and high prevalence areas in California. They make a point that individuals with positive / equivocal result were usually non-Hispanic however one would assume this based on the lower prevalence of Hispanics in this cohort. The seroprevalence to C6 peptide should be assessed directly in Hispanics vs non-Hispanics taking into account the sample size for each.

4) The authors selected randomly 9 negative samples by standalone C6 ELISA, two thirds of which were female. What is the rationale for selecting 9 samples (vs more) and why were they not gender matched with the C6 seropositive populations?

5) Table 2 should include a row that lists the C6 findings for all 1700 patients. It should also include data for all 941 samples from high endemic area, as well as for 759 patients from low enedemic area; followed by the data for each county in that area. The data need to be clearly labeled (what are the data in parenthesis? Do the data represent positive results?). It would help to present the number of positives in addition to percentages.

6) The paragraph on B. miyamotoi GlpQ seroreactivity discusses seroprevalence in high vs low endemic regions for B. burgdorferi. This discussion is based on only two positive samples, and it is difficult to conclude much in the difference in prevalence between the two regions, except that the seroprevalence is relatively low. One could also postulate that the infection could have been acquired elsewhere and not in the counties/state where the samples were obtained.

7) A similar comment holds true for the paragraph on B. burgdorferi IgG western blot. The authors provide percentages and CI for seroprevalence in Marin county, but only one donor was seropositive there. With such small numbers it is difficult to assess the significance of seroprevalence and one should be careful to not overinterpret the data. For example, is it fair to state that San Luis Obispo county had the highest prevalence in the low endemic area since it had the only positive case (out of 759) in this region?

8) Line 369 in the Discussion: is it low specificity or high sensitivity of C6 ELISA that would over-estimate true exposure?

Reviewer #2: This is a very well written paper from several experienced investigators. They report a comprehensive sero-survey of B. burgdorferi and B. miyamotoi in counties of high and low endemicity for LD in California. The study design is straightforward and the results confirm previous observations of low prevalence of LD in CA (<0.5%). They found an even lower prevalence of antibodies to B. miyamotoi (0.1%). It is interesting that they found that the two positive miyamotoi cases were positive in C6 and Qlp. The following paper should be discussed: Borrelia miyamotoi infection leads to cross-reactive antibodies to the C6 peptide in mice and men. Koetsveld J, Platonov AE, Kuleshov K, Wagemakers A, Hoornstra D, Ang W, Szekeres S, van Duijvendijk GLA, Fikrig E, Embers ME, Sprong H, Hovius JW. Clin Microbiol Infect. 2020 Apr;26(4):513.e1-513.e6. doi: 10.1016/j.cmi.2019.07.026. Epub 2019 Aug 9. PMID: 31404672

The results are well discussed and limitations of the study have be identified. I have no further major or minor suggestions to improve this paper.

6. PLOS authors have the option to publish the peer review history of their article (what does this mean?). If published, this will include your full peer review and any attached files.

Reviewer #1: No

Reviewer #2: No

---

## [Author Response · Author response to Decision Letter 0]

19 Nov 2020

Manuscript Name: Borrelia burgdorferi and Borrelia miyamotoi Seroprevalence in California Blood Donors

Response to Reviewers

Reviewer 1: 

Comment 1) Figure 1 should include the number of samples that came from each county as well as the percentage. Figure Legends should be provided. 

Response to Comment #1: The California county map (figure 1) now includes both the number and percent of the sera samples from each county sampled. The figure legends have been provided in line with the text per instructions to authors. Figure 1 and legend to figure 1 starts at line 117. Figure 2 and legend to figure 2 starts at line 157 and figure 3 and legend to figure 3 starts at line 311.

Comment 2) The 2nd paragraph of the Results (C6 ELISA Screen) provides information on how the 91 patients were selected for downstream analyses, but this information is not presented anywhere. My suggestion is to present this in a form of a Table or a flow chart since all subsequent analyses in the manuscript stem from this selection/screen.

Response to Comment #2: The authors appreciate this comment. A flow chart has been added (figure 2) displaying the process of testing strategies along with the number of samples included for each test. Figure 2 and legend start at line 157. See separate figure 2 uploaded in Plos One submission website 

Comment 3) In the second paragraph of the C6 ELISA Screen section, the authors discuss the distribution of Hispanics and non-Hispanics among low and high prevalence areas in California. They make a point that individuals with positive / equivocal result were usually non-Hispanic however one would assume this based on the lower prevalence of Hispanics in this cohort. The seroprevalence to C6 peptide should be assessed directly in Hispanics vs non-Hispanics taking into account the sample size for each.

Response to Comment #3: The authors appreciate this comment. We have re-worded the results of the C6 ELISA in terms of ethnicity. The paragraph starts at line 278. To address the small cell counts in ethnicity, a Fisher’s exact test was performed, which does take into account the sample sizes for each group. 

Comment 4) The authors selected randomly 9 negative samples by standalone C6 ELISA, two thirds of which were female. What is the rationale for selecting 9 samples (vs more) and why were they not gender matched with the C6 seropositive populations?

Response to Comment #4: The authors thank the reviewer for this insight. We selected only 9 negative samples and not more due to resource limitations of second tier testing. Though the samples could have been matched on gender, given the small number of negative samples we could afford, the authors chose to balance by endemnicity status. Ultimately there was no difference in seropositivity between males and females, so the authors do not feel that the 2/3 bias towards females in the negative samples compromises the study. Nonetheless we noted this bias in the limitations paragraph starting at line number 491.

Comment 5) Table 2 should include a row that lists the C6 findings for all 1700 patients. It should also include data for all 941 samples from high endemic area, as well as for 759 patients from low endemic area; followed by the data for each county in that area. The data need to be clearly labeled (what are the data in parenthesis? Do the data represent positive results?). It would help to present the number of positives in addition to percentages. 

Response to Comment #5: We have revised table 2 and added the findings from the C6 ELISA, which was used as a screening test. The data are clearly labeled and are presented as the number positive (% positive). Table 2 is on line 252. 

Comment 6) The paragraph on B. miyamotoi GlpQ seroreactivity discusses seroprevalence in high vs low endemic regions for B. burgdorferi. This discussion is based on only two positive samples, and it is difficult to conclude much in the difference in prevalence between the two regions, except that the seroprevalence is relatively low. One could also postulate that the infection could have been acquired elsewhere and not in the counties/state where the samples were obtained. 

Response to Comment #6: The authors agree that two positive samples limit detailed discussion on regional differences. We deleted line the sentence starting on line 288 that states that there was significant difference between high and low endemic counties. Beginning on line 417 we mentioned that seropositivity could have been acquired outside county of residence

Comment 7) A similar comment holds true for the paragraph on B. burgdorferi IgG western blot. The authors provide percentages and CI for seroprevalence in Marin County, but only one donor was seropositive there. With such small numbers it is difficult to assess the significance of seroprevalence and one should be careful to not over interpret the data. For example, is it fair to state that San Luis Obispo county had the highest prevalence in the low endemic area since it had the only positive case (out of 759) in this region?

Response to Comment #7: The sentence comparing Marin County seroprevalence to San Luis Obispo County seroprevalence was deleted. We substituted text on the overall seroprevalence along with its exact 95% CI were presented on all IgG western blot for all B. burgdorferi seropositive samples from each of the high endemic counties as well as the one positive B. burgdorferi from the low endemic county for LD, starting on line 308.

Comment 8) Line 369 in the Discussion: is it low specificity or high sensitivity of C6 ELISA that would over-estimate true exposure?

Response to Comment #8: The authors agree that high sensitivity and low specificity can over- estimate true exposure of LD, especially in a low prevalence state. Wormser et al, 2013 addressed this issue in his paper (reference 49): Single-tier testing with the C6 peptide ELISA kit was compared with two-tier testing for Lyme disease and the authors concluded that the C6 ELISA as a “single-step serodiagnostic test provided increased sensitivity in early Lyme disease with comparable sensitivity in later manifestations of Lyme disease. The C6 ELISA had slightly decreased specificity”. See revisions on manuscript starting on line 456 citing increased sensitivity and decreased specificity as potential cause of over-estimation. 

Reviewer #2: 

Comment 1) This is a very well written paper from several experienced investigators. They report a comprehensive sero-survey of B. burgdorferi and B. miyamotoi in counties of high and low endemicity for LD in California. The study design is straightforward and the results confirm previous observations of low prevalence of LD in CA (<0.5%). They found an even lower prevalence of antibodies to B. miyamotoi (0.1%). It is interesting that they found that the two positive miyamotoi cases were positive in C6 and Qlp. The following paper should be discussed: Borrelia miyamotoi infection leads to cross-reactive antibodies to the C6 peptide in mice and men. Koetsveld J, Platonov AE, Kuleshov K, Wagemakers A, Hoornstra D, Ang W, Szekeres S, van Duijvendijk GLA, Fikrig E, Embers ME, Sprong H, Hovius JW. Clin Microbiol Infect. 2020 Apr;26(4):513.e1-513.e6. doi: 10.1016/j.cmi.2019.07.026. Epub 2019 Aug 9. PMID: 31404672

The results are well discussed and limitations of the study have be identified. I have no further major or minor suggestions to improve this paper.

Response to Comment #1: Thank you for your suggestion on an article that should be discussed in this manuscript. The article has been discussed starting at line 386.

---

## [Decision Letter · Decision Letter 1]

1 Dec 2020

Borrelia burgdorferi and Borrelia miyamotoi Seroprevalence in California Blood Donors

PONE-D-20-23819R1

Dear Mrs. Brummitt,

We’re pleased to inform you that your manuscript has been judged scientifically suitable for publication and will be formally accepted for publication once it meets all outstanding technical requirements.

Kind regards,

R. Mark Wooten, Ph.D.

Academic Editor

PLOS ONE

Additional Editor Comments (optional):

Reviewers' comments:

Reviewer's Responses to Questions

**Comments to the Author**

1. If the authors have adequately addressed your comments raised in a previous round of review and you feel that this manuscript is now acceptable for publication, you may indicate that here to bypass the “Comments to the Author” section, enter your conflict of interest statement in the “Confidential to Editor” section, and submit your "Accept" recommendation.

Reviewer #1: All comments have been addressed

Reviewer #2: All comments have been addressed

2. Is the manuscript technically sound, and do the data support the conclusions?

Reviewer #1: Yes

Reviewer #2: Yes

3. Has the statistical analysis been performed appropriately and rigorously? 

Reviewer #1: Yes

Reviewer #2: Yes

4. Have the authors made all data underlying the findings in their manuscript fully available?

Reviewer #1: Yes

Reviewer #2: Yes

5. Is the manuscript presented in an intelligible fashion and written in standard English?

Reviewer #1: Yes

Reviewer #2: Yes

6. Review Comments to the Author

Reviewer #1: The revised manuscript addresses the reviewer’s concerns. It brings to light important issue regarding seroprevalence of B. burgdorferi and B. miyamotoi in California which caries public health implications.

Reviewer #2: (No Response)

7. PLOS authors have the option to publish the peer review history of their article (what does this mean?). If published, this will include your full peer review and any attached files.

Reviewer #1: No

Reviewer #2: No

---

## [Editor Report · Acceptance letter]

7 Dec 2020

PONE-D-20-23819R1 

*Borrelia burgdorferi* and *Borrelia miyamotoi* Seroprevalence in California Blood Donors 

Dear Dr. Brummitt:

I'm pleased to inform you that your manuscript has been deemed suitable for publication in PLOS ONE. Congratulations! Your manuscript is now with our production department. 

Kind regards, 

on behalf of

Dr. R. Mark Wooten 

Academic Editor

PLOS ONE